# Elemental Chemometrics as Tools to Depict Stalked Barnacle (*Pollicipes pollicipes*) Harvest Locations and Food Safety

**DOI:** 10.3390/molecules27041298

**Published:** 2022-02-15

**Authors:** Bernardo Duarte, Renato Mamede, Irina A. Duarte, Isabel Caçador, Susanne E. Tanner, Marisa Silva, David Jacinto, Teresa Cruz, Vanessa F. Fonseca

**Affiliations:** 1MARE—Marine and Environmental Sciences Centre, Faculdade de Ciências, Universidade de Lisboa, Campo Grande, 1749-016 Lisbon, Portugal; rjmamede@fc.ul.pt (R.M.); iaduarte@fc.ul.pt (I.A.D.); micacador@fc.ul.pt (I.C.); setanner@fc.ul.pt (S.E.T.); mpdsilva@fc.ul.pt (M.S.); vffonseca@fc.ul.pt (V.F.F.); 2Departamento de Biologia Vegetal, Faculdade de Ciências, Universidade de Lisboa, Campo Grande, 1749-016 Lisboa, Portugal; 3Departamento de Biologia Animal, Faculdade de Ciências, Universidade de Lisboa, Campo Grande, 1749-016 Lisboa, Portugal; 4MARE—Marine and Environmental Sciences Centre, Laboratório de Ciências do Mar, Universidade de Évora, 7521-903 Sines, Portugal; djacinto@uevora.pt (D.J.); tcruz@uevora.pt (T.C.); 5Departamento de Biologia, Escola de Ciências e Tecnologia, Universidade de Évora, 7002-554 Évora, Portugal

**Keywords:** seafood, chemometrics, traceability, food safety, multi-elemental signature, barnacles

## Abstract

The stalked barnacle *Pollicipes pollicipes* is an abundant species on the very exposed rocky shore habitats of the Spanish and Portuguese coasts, constituting also an important economical resource, as a seafood item with high commercial value. Twenty-four elements were measured by untargeted total reflection X-ray fluorescence spectroscopy (TXRF) in the edible peduncle of stalked barnacles sampled in six sites along the Portuguese western coast, comprising a total of 90 individuals. The elemental profile of 90 individuals originated from several geographical sites (*N* = 15 per site), were analysed using several chemometric multivariate approaches (variable in importance partial least square discriminant analysis (VIP-PLS-DA), stepwise linear discriminant analysis (S-LDA), linear discriminant analysis (LDA), random forests (RF) and canonical analysis of principal components (CAP)), to evaluate the ability of each approach to trace the geographical origin of the animals collected. As a suspension feeder, this species introduces a high degree of background noise, leading to a comparatively lower classification of the chemometric approaches based on the complete elemental profile of the peduncle (canonical analysis of principal components and linear discriminant analysis). The application of variable selection approaches such as the VIP-PLS-DA and S-LDA significantly increased the classification accuracy (77.8% and 84.4%, respectively) of the samples according to their harvesting area, while reducing the number of elements needed for this classification, and thus the background noise. Moreover, the selected elements are similar to those selected by other random and non-random approaches, reinforcing the reliability of this selection. This untargeted analytical procedure also allowed to depict the degree of risk, in terms of human consumption of these animals, highlighting the geographical areas where these delicacies presented lower values for critical elements compared to the standard thresholds for human consumption.

## 1. Introduction

Seafood is a key component of the human diet, with increasing demands nowadays [1]. In the last decades, seafood comprised about 16.7% of animal protein intake per capita, with a tendency to increase due to a higher concern with a healthier food regime (world per capita consumption on average in the 1960s: 9.9 kg, and in 2013: 19.7 kg) [1]. With higher demand, there is also an increased risk of seafood mislabelling, either unintentional or with the intent to gain profit from illegal practices [2]. Recent fisheries expansion and globalization, along with greater public awareness regarding food quality, led to a growing interest in several issues related to seafood authenticity and compliance with food legislation. As per the European Regulation No 1379/2013 [3] fishery and aquaculture products must be labelled with the commercial designation, species scientific name, production method (e.g., caught, farmed), fishing gear (e.g., hook, trap, trawl), and catch or production area. Errors in label information about seafood origin and production process are increasing in frequency to such an extent that seafood products are nowadays the second category of food most vulnerable to fraud [1]. The implementation of seafood labelling and traceability all over the world led to an increasing number of studies aiming to develop biochemical and elemental tags to provide unequivocal natural labels of the product capture or production, independently of the producer’s information [4]. These tools comprise DNA barcoding techniques [5,6], fatty acid profiling [7,8], microbial profiling [9] and trace element fingerprinting or profiling [10,11,12,13,14,15]. Considering elemental profiling, total X-ray fluorescence (TXRF) spectroscopy has been pointed out as a high-throughput non-targeted analytical tool to efficiently quantify the elemental composition of seafood products, tracing the product geographical origin whilst simultaneously, if the targeted tissue is edible, providing accurate food safety values [13,16,17,18].

Barnacles are key organisms on rocky shores, being exploited as important economic resources, namely the giant barnacle *Austromegabalanus psittacus* in Chile, the acorn barnacle *Megabalanus azoricus* in Azores (Portugal) and the stalked barnacles of the genus *Pollicipes* [19]. Considering this last genus, all species are edible (*P. polymerus*, north-eastern Pacific Ocean, *P. elegans*, tropical eastern Pacific Ocean, *P. caboverdensis*, Cape Verde Islands and *P. pollicipes,* north-eastern Atlantic Ocean), but *P. pollicipes* can be considered the most important intertidal economical resource on rocky shores of North Spain and continental Portugal, with a commercial value ranging from EUR 20 to 200 per kg in restaurants [20]. In terms of species conservation, several legislations are enforced in Portugal aiming to establish limits for barnacle harvesting and geographical areas where this activity is allowed, either for professional or recreational harvesting [21,22,23]. Nevertheless, the identification of the harvesting sites is based on the information provided by the fishermen [21], and thus more prone to mislabelling events. Previous works [10] have identified the capitula elemental signatures as potential tracers for *P. pollicipes* harvesting origin, with an 87% success rate. Nevertheless, the capitula is not the edible part of the organism. Thus, an approach focusing on the edible peduncle would allow not only to develop traceability tools, based on chemometric approaches, but also provide a risk assessment of the edible parts for human consumption [13].

Accordingly, the present work aims to disclose the potential of TXRF-based non-targeted multi-elemental fingerprinting of *P. pollicipes* peduncle using several chemometric approaches for geographical origin traceability assessment, with simultaneous evaluation of food safety values related to the presence and concentration of potentially toxic elements. This represents a rather novel approach to the traceability of these high commercial-value organisms, not only due to the application of a non-targeted analytical tool (TXRF) but also due to the application of several chemometric approaches to the edible peduncle of *P. pollicipes*, in opposition to previous works that focused on the calcarean structures of these organisms [10].

## 2. Results

### 2.1. Barnacle Peduncle Elemental Concentrations and Safety Values

Arsenic concentrations in *P. pollicipes* peduncle ranged from 0 to 34.6 mg/kg, with the lowest concentrations found in the individuals captured at the northern location, Viana do Castelo (Figure 1A). A similar pattern was found in the Br peduncle content, ranging in this case from 1.4–70.8 mg/kg (Figure 1B). Regarding Ca and Cl contents (Figure 1C,D, respectively), these were also found to be lower in the animals collected at Viana do Castelo, followed by *P. pollicipes* individuals collected at Matosinhos and Sagres, with the concentrations of these elements ranging from 47.6 to 3335.4 mg/kg and from 13.7 to 2877.0 mg/kg peduncle wet weight, respectively. As for potassium peduncle content (Figure 1E; ranging 9719.1 to 1138.3 mg/kg), it was higher in individuals collected in both northern sampling areas (Viana do Castelo and Matosinhos) when compared to the lowest values observed in the animals harvested at the southern locations (Sines and Sagres). The lowest sodium contents (1218.1 mg/kg) were found in individuals captured at Viana do Castelo, while the animals from Ericeira and Sines exhibited the highest contents of this element (maximum = 9220.7 mg/kg) (Figure 1F). Phosphorous peduncle contents ranged from 437.0 to 3686.4 mg/kg and were found to have their lowest values in the individuals collected at Sines, Sagres and Ericeira (Figure 1G). Praseodymium showed a highly conservative behaviour along the barnacle populations surveyed, with no significant differences observed between the sampling sites and ranging from 0 to 4.7 mg/kg (Figure 1H). Rubidium peduncle content (Figure 1I) was also found to be very stable along the sampling gradient with lower values detected in the animals collected at Viana do Castelo, Matosinhos, Sines and Sagres, when compared to the ones harvested at Ericeira and Cabo Raso (ranging from 0 to 7.8 mg/kg). Ruthenium peduncle content was very stable along the individuals surveyed from all sampling sites, ranging from 0 to 73.8 mg/kg (Figure 1J). Sulphur content in the peduncle tissues of the surveyed barnacles (Figure 1K), was found to be rather stable along the surveyed areas, with significantly lower contents in the animals collected at Viana do Castelo (overall range = 769.6 to 7331.4 mg/kg). Selenium peduncle content (Figure 1L) was found to be lowest in the animals captured at Sines and Viana do Castelo (minimum value = 0.16 mg/kg), while the individuals harvested at Matosinhos and Ericeira showed the higher values of this element (maximum value = 0.74 mg/kg). Samarium peduncle contents (Figure 1M) showed higher values in the animals harvested at Viana do Castelo and Matosinhos, while the remaining surveyed areas showed significantly lower values (overall ranging from 0 to 3.0 mg/kg). Similar to what was found for other elements, strontium values were also found to have lower values in the animals captured at Viana do Castelo (minimum value = 0.12 mg/kg), being very similar in the animals collected at the remaining sampling sites, ranging to a maximum of 83.1 mg Sr/kg detected in the individuals collected at Cabo Raso (Figure 1N). Regarding titanium and yttrium peduncle contents no significant differences could be observed among the individuals collected at the six surveyed sites (Figure 1O and Figure 1Q, respectively). Lastly, vanadium peduncle contents (Figure 1P) showed the highest values in the individuals collected at Cabo Raso and Sines (maximum value = 1.0 mg/kg), being this element present in very low concentrations in the individuals collected at the remaining sampling sites.

The animals harvested at Viana do Castelo and Matosinhos showed the lowest peduncle concentrations of Cr. Maximum values of this element were detected in the samples collected at Ericeira, Sines and Sagres (Cr concentration range: 0–2 mg/kg) (Appendix A). Nevertheless, and considering the international guidelines thresholds for this element (15 mg/kg ww), all the analysed samples showed safe values which concerns Cr concentration (Figure 2A, ratio bellow 1). Copper values ranged from 0.15 to 1.8 mg/kg, being the lowest values observed in the animals captured at Viana do Castelo and Matosinhos, and the highest at Ericeira and Cabo Raso (Appendix A). Again, none of the samples surpassed the safety thresholds for human consumption regarding this element (Figure 2B, ratio bellow 1). Regarding the Fe peduncle contents, except for samples collected at Matosinhos, all sampling sites exhibited individuals with iron contents above the recommended thresholds for human consumption (Figure 2C). To be more precise, 42 barnacle samples (47%) surpassed the 43 mg/kg threshold (ratio above 1). The iron values in the surveyed samples ranged from 2.4 to 211.4 mg/kg, with the highest Fe values detected in the samples collected at Ericeira, Cabo Raso and Sagres (Appendix A). About 70% of the samples surpassed the regulatory Mn threshold of 1 mg/kg (Figure 2D), with Mn concentrations ranging from 0 to 9.3 mg/kg over the surveyed sites (Appendix A). The lowest values of this element were detected in the peduncle of the animals captured at Matosinhos, while no significant differences could be observed among the animals collected at the remaining sites. Nickel values (0–0.9 mg/kg, Appendix A) were found to be below the regulatory thresholds (80 mg/kg), with minimum values assessed for the individuals collected at Viana do Castelo and maximum values in the animals captured at Ericeira, Sines and Sagres (Figure 2E). Lead concentration regulatory thresholds (1 mg/kg) was surpassed in 30% of the barnacle samples (Figure 2F). Only the samples collected at Viana do Castelo were below this threshold value, being significantly lower than the observed in individuals harvested at Ericeira, Cabo Raso, Sines (maximum detected values of 14.9 mg/kg) and Sagres (Appendix A). Regarding Zn peduncle concentrations (Appendix A) these were found to be minimum in the animals collected at Viana do Castelo and Matosinhos (minimum value = 3.5 mg/kg), being this also the only site where none of the samples surpassed the regulatory 40 mg/kg threshold (Figure 2G). Maximum Zn values (571.0 mg/kg) were detected in the samples collected at Ericeira and Cabo Raso. Accordingly, 76% of the samples surpassed the Zn regulatory thresholds for human consumption.

Analysing the concentration trends of the detected elements (Figure 3), it can be observed that most of the elements show significant and positive correlations between them, with only a few exceptions such as potassium, praseodymium and samarium. These show a large number of inverse significant correlations with the remaining analysed elements. This introduces a high degree of multicollinearity, that will latterly be discussed in terms of the chemometric approach.

### 2.2. Chemometric Provenance Classification

Several multivariate statistical analyses were used as chemometric tools to depict the geographical elemental signatures of the animals harvested in the different locations. For visualization purposes, the resulting plots from the LDA, CAP, PLS-DA and RF analyses are presented in Figure 4. In all projections, the same pattern is observed with three geographical groups or areas formed by distinct sampling sites: (i) a northern area composed of the samples collected at Viana do Castelo and Matosinhos; (ii) a central area gathering the samples harvested at Ericeira and Cabo Raso and (iii) a southern area composed by the samples collected at Sines and Sagres. Observing the confusion matrixes (Table 1) from the different tested chemometric approaches it is also possible to observe that most of the misclassification events occur within these three groups. In terms of models’ accuracy, the stepwise linear discriminant analysis (S-LDA) showed the highest accuracy in classifying the samples correctly according to their geographical origin (84.4%), based on the elemental concentrations of As, Se, Ca, Fe, Zn, V, Ni, P and Sm as descriptors. This is an evident improvement in comparison with the use of the whole elemental dataset within a similar technique, as observed in the accuracy results (72.7%) for the stepwise linear discriminant analysis (LDA). Comparing the results from the partial least-squares discriminant analysis (PLS-DA) and the variable importance in projection partial least-squares discriminant analysis (VIP-PLS-DA), no improvements in terms of classification accuracy could be observed (77.8% accuracy in both methods, Table 1). In this last model, only the variables with a variable importance in projection (VIP) score above 1 were considered (in this case Br, Ca, Cl, Fe, K, Na, Ni, S, Se, Sr and Zn). Canonical analysis of principal (CAP) components and random forest (RF) analysis had the lowest classification accuracy of 73.3% and 70%, respectively. In terms of sensitivity, S-LDA showed the highest sensitivity and specificity, followed by both PLS-DA approaches (PLS-DA and VIP-PLS-DA).

Analysing the element relative weight for models’ accuracy and sample geographical classification, there were some consistent patterns among the tested chemometric approaches (Figure 5). Selenium is one of the most relevant elements for correct sample classification in all the tested approaches. Sodium also presented relevant importance for the classification accuracy of four models, with the exception of the LDA (Figure 5C) and S-LDA (Figure 5D) approaches. Apart from the S-LDA approach, Ni also showed a high degree of importance in the remaining chemometric models applied. Calcium and Zn showed a preponderant role to achieve high accuracy classifications when applying RF and CAP methodologies (Figure 5E,F). Arsenic was found to be highly relevant for achieving the high accuracy values observed in the S-LDA approach. Regarding the remaining elements analysed, these were found to have very similar weights within the variables tested in the different tested approaches.

## 3. Discussion

Animal tissues elemental composition is intrinsically linked to the chemical composition of their environment, but it is also modulated by the animal physiology and food sources intake [24,25,26]. To our knowledge, the elemental signature of barnacles’ edible peduncle has not been used for geographical traceability purposes, with only one study focusing on *P. pollicipes* calcified capitula structures for this purpose [10]. Given that *P. pollicipes* is considered the most important intertidal economical resource on rocky shores of North Spain and continental Portugal, due to their high commercial value [27], the need to develop tools that allow tracing the harvesting site of the commercialized barnacles is greatly increased. Additionally, if this approach can simultaneously focus on the geographical provenance of the product while evaluating its food safety requirements for human consumption, its relevance is reinforced.

Since there is no specific information available in the literature concerning the best elemental data set to discriminate the geographical origin of *P. pollicipes* based on its edible peduncle, the first chemometric analysis focused on all the 24 elements measured, by exploring the data employing LDA and CAP are reported in the present study. The accuracy of both approaches was not the highest among the tested models. The high multicollinearity between elemental variables as well as geographical-unrelated variability is often considered a source of increasing noise, thus reducing the maximum accuracy of whole dataset based models [28]. This was also observed in the present study, with a high number of elements presenting significant correlations between each other, increasing the multicollinearity of the dataset. The main significant patterns of covariations were found among between Zn and Na and, in particular, between the halogens Br and Cl (ρ > 0.78), followed by correlations between Na, Zn, Br, Sr and Ni (ρ > 0.70). These are not only the most abundant elements in marine water, but the concentration of these elements in seawater is rather conservative, exhibiting very low variation [29]. Due to their filter and captorial feeding habits, barnacles not only capture preys with their cirri [30] but also filtrate seawater, and thus the high abundance of these elements allied to the common uptake rates supports the high multicollinearity observed. This might explain the high concentration of these elements measured in the barnacles’ peduncles. On the opposite end, praseodymium, samarium and yttrium concentrations in the barnacles’ peduncles showed several inverse correlations with most of the analysed elements. These elements are among the less abundant (yttrium is in fact considered a Rare Earth Element, REE), not only in seawater but also in the whole planet, thus suggesting a different uptake process from the highly abundant elements. Moreover, these elements show a similar accumulation behaviour to the observed for K, with inverse correlations with Na. Marine organisms live in a medium with an inexhaustible supply of K, being the uptake of these elements often intrinsically regulated by antiport systems [31]. In the case of praseodymium, samarium and yttrium, their rare character indicates that their uptake mechanism is not the same as for K, but the results here presented point to possible co-transport systems.

Thus, the reduction of explanatory variables is often an efficient way to reduce this background noise, especially given a future practical implementation of the methodology. Variable selection is highly dependent on the sample matrix properties as well as on the dataset overall characteristics [28]. Particularly, in the case of seafood samples, this is highly dependent on the animals feeding, moving and ecological traits, as well as the variability of the marine habitats, that overall affect the animal element uptake [24,25,26] and thus the selected variables. This points to a case-specific selection of variables and not to a transversal elemental selection, independently of the samples. For these purpose, two approaches based on variable selection were undertaken VIP-PLS-DA and S-LDA. The latter presented the highest degree of accuracy, as well as of all the other model performance metrics considered, namely precision, sensitivity and specificity.

Akin to the CAP analyses, S-LDA is probably the most frequently used classification technique when dealing with elemental profiling data [28]. Some critics of this approach are due to the randomness of the variable selection, rather than an effective significance of the variables extracted [32]. Nevertheless, if this approach is compared with other variable selection models with a high degree of randomness (like Random Forests) and with a low degree of randomness, such as VIP-PLS-DA, some conclusions on the selected variables can be drawn, especially for those commonly selected among random and non-random approaches. In terms of variable selection, VIP-PLS-DA consists of a more robust and flexible algorithm, particularly fit for the classification of a large number of samples, and is recommended as a more powerful tool for reliable variable selection compared to S-LDA [33]. Although some studies point to some disadvantages of PLS-DA, when used alone (for e.g., problems of overfitting and PLS scores plots), this integration of VIPS-PLS-DA within a full classification procedure is advised [34]. In fact, as discussed below, several of the selected variables, as key drivers of the PLS-DA grouping, were simultaneously highlighted by the remaining chemometric approaches tested, reinforcing its integrated use along with other multivariate approaches [34].

Comparing the elements selected by the VIP-PLS-DA (VIP score > 1) with the elements selected by the S-LDA, it was possible to observe that from the 8 elements selected in the latter (As, Se, Zn, Ca, V, Fe, Ni and P), 63% (Se, Zn, Ca, Fe and Ni) were also selected by the VIP-PLS-DA, a method with a lower degree of randomness [33]. Even when comparing the VIP-PLS-DA most relevant elements with a random approach, such as the one provided by random forests, it is possible to observe that the eight most important variables (Se, Zn, Ca, Cl, Fe, Sr, Na and Ni) are common in both approaches. Comparing RF with S-LDA, several elements are also common to both approaches considering the most relevant elements for the high accuracy values observed (Se, Zn, Ca, Fe and Ni). These facts point to a reliable selection of variables obtained by the S-LDA approach, which in the present study gathered the highest degree of accuracy [33]. Among these elements are not only highly abundant elements, such as Zn, Ca, Fe and P but are also elements whose concentrations in the environment and in the barnacle tissues are far lower (As, Se, V and Ni), excluding any environmental prevalence effect on the variable importance to the overall method accuracy. Comparing to previous traceability works focusing on this species [10], the selected elements by the presented chemometric approaches differ greatly. Albuquerque et al. (2016) used the calcified capitula as an analytical matrix, which revealed that Ba, B, Cd, Cr, Li, Mg, Mn, P, Pb, Sr and Zn were the most important elements for discriminating the individuals’ sampling areas. This is likely due to the different analytical techniques (ICP-MS versus TXRF) and choice of matrix (capitula versus edible soft tissues), as previously observed for other organisms using calcified structures [35]. Notably, some of the observed variability within sites (resulting in a cluster dispersion in LDA and CAP plot in some groups), can be due to different reproductive stages of the individuals (differing stages of ovarium development within the edible peduncle) [36], since collection was performed during the reproductive season. Another potential influencing factor is the small-scale variation of the habitat in which the animals were collected, as it was previously observed for this species using the calcified capitula [10] as well as for other organisms and their calcified structures [15].

By comparison to other seafood products and their elemental profiles available in the literature, some of the most discriminant elements found in the present study might be linked to anthropogenic pollution (As, Se, V, Ni) [13,28,37]. Both the individual element concentrations in the barnacle peduncle and the multivariate projections (LDA and CAP) seem to indicate three distinct geographical groups or areas (north, centre and south), that can be attributed to a latitudinal seawater chemical gradient alongside different degrees of anthropogenic pressures [38]. Moreover, these elemental accumulation patterns are in line with previous studies with the stalked barnacle *P. pollicipes* along the Portuguese coast [39,40]. In these previous works, the results for some of the legislated elements were also found to be above the international thresholds for human consumption, and in some cases with values above the ones here reported [39,40]. Nevertheless, it should be taken into account that only for Pb, these thresholds are differentially established for crustaceans and fishes. For the remaining elements, a unique threshold for seafood is available, without taking into account the normally higher value present in deposit and suspension feeder organisms, thus leading to several samples from the present study being above the recommended threshold for human consumption. Additionally, in Portugal, stalked barnacles are generally not consumed as the main dish, and thus, it is not expectable to constitute a large portion of the daily intake food amount. In this way, the risks inherent to barnacle consumption are reduced considering the above threshold concentration values, observed for some elements and some barnacle populations along the Portuguese coast.

The application of untargeted X-ray fluorescence spectroscopy analysis has been growing for traceability studies based on elemental fingerprinting [13,17,18], producing high-throughput results and a large number of elements, analysed with minimum sample preparation [41]. In the present study, this high-throughput dataset appears also to be suitable for chemometric approaches with a simultaneous evaluation of the food safety risk of stalked barnacles, being, therefore, a promising technique, equally efficient when compared to the more classic targeted techniques, such as inductively coupled plasma-based (ICP-MS) analysis.

## 4. Materials and Methods

### 4.1. Sample Collection

Stalked barnacles (*Pollicipes pollicipes*) were collected in six rocky shore sites, along the Western Portuguese coast (Figure 6), in July and August 2021. A total of 90 *P. pollicipes* individuals were collected in all sampling areas. Individuals were transported fresh to the laboratory. For morphometry purposes, 15 individuals per site were weighed (mean ± standard deviation, 3.9 ± 1.6 g) and their total height (mean ± standard deviation, 45.9 ± 11.6 mm) measured. No significant differences were observed among sites for both weight and height (*p* > 0.05). After all morphometric measurements, the individuals were dissected by separating the edible part of the peduncle from the rest of the organism, by removing the outer cuticle. Peduncle samples were freeze-dried and stored at −80 °C until further processing.

### 4.2. Total Elemental Fingerprinting

All labware used for elemental analysis were decontaminated in acid baths for 48 h before use. Freeze-dried samples (15 replicates per site) were mineralized with HNO_3_ in Teflon reactors, following a microwave digestion process (Multiwave GO, Anton Paar GmbH, Graz, Austria) according to the EPA 3052 method [42]. After cooling, an internal standard (Gallium) was added to each sample, and 5 μL of each sample was then applied to a siliconized quartz disk (BruckerNano, Germany) and dried. Elemental concentrations (As, Br, Ca, Cl, Cr, Cu, Fe, K, Mn, Na, Ni, P, Pb, Pr, Rb, Ru, S, Se, Sm, Sr, Ti, V, Y, Zn) were determined by total reflection X-ray fluorescence spectroscopy (TXRF S2 PICOFOX, Brucker, Germany). Instrumental recalibration (gain correction, sensitivity analysis and multi-elemental standards) and analytical blanks were used for quality control. Elemental concentrations were determined by comparison with the internal standard [13]. Extraction efficiency was confirmed through the analysis of International certified reference materials (ERM-CE278k Mussel Muscle, Table 2).

For food safety purposes, element concentrations were compared against available regulatory safety values, specifically for Pb (0.20 mg/kg), Cu (30 mg/kg), Zn (40 mg/kg), Mn (1 mg/kg), Ni (80 mg/kg), Fe (43 mg/kg) and Cr (15 mg/kg), according to international guidelines [43,44,45]. For human consumption suitability assessment, a ratio between the individual measured elemental value and the established threshold for the same element was calculated. Ratio values above one indicate elemental concentrations above the recommended international thresholds for human consumption.

### 4.3. Statistical and Chemometric Analysis

Non-parametric Kruskal–Wallis with Bonferroni post hoc test was used for pairwise comparisons between the elemental concentrations of the individuals collected at the different geographical areas and were performed in R-Studio Version 1.4.1717 using the *agricolae* package [46]. Boxplots with probability density of the data at different values smoothed by a kernel density estimator were computed and plotted using *ggplot2* [47] package in R-Studio Version 1.4.1717. Spearman correlation coefficients and statistical significance among elements concentration in the peduncle of the analysed individuals were computed using the *corrplot* [48] package in R-Studio Version 1.4.1717.

For the chemometric approach, five multivariate statistical methodologies were employed. The Primer 6 (version 6.1.13, with PERMANOVA + addon version 1.0.3) software [49] was used to carry out multivariate statistical analyses using non-parametric multivariate analysis packages. Following data transformation (log x + 1) and dispersion weighting, the resemblance matrix of all variables (based on Euclidean distances) was analysed using canonical analysis of principal coordinates (CAP), in order to evaluate the ability to successfully classify individuals to their collection areas. This multivariate approach is insensitive to heterogeneous data and frequently used to compare different sample groups using the intrinsic characteristics (elemental concentrations) of each group [13,50]. Variables’ relative importance was calculated by dividing the maximum absolute Spearman correlation coefficient of each variable with the canonical axis CAP1 and CAP2 by the sum of all variables’ maximum absolute Spearman correlation coefficients with the canonical axis CAP1 and CAP2.

Partial least-squares discriminant analysis (PLS-DA) and variable importance in projection partial least-squares discriminant analysis (VIP-PLS-DA) were performed using the *DiscriMiner* package [51] in R-Studio Version 1.4.1717. For VIP-PLS-DA analysis, only variables with a VIP score > 1 were included. Variable relative importance was calculated by dividing the VIP score of each variable by the sum of all VIP scores.

Linear discriminant analysis (LDA) was computed using the *MASS* package [52] in R-Studio Version 1.4.1717. For LDA, normality of the data in each class, as well as covariance between sample matrixes, were ensured. The complete sample data was divided into 40% of the samples data set for model training, and the remaining 60% of the samples data set was used for the model test. Variable relative importance was calculated by dividing the sum of the absolute value of each variable first two linear discriminants with the whole variable dataset sum of the absolute value of the first two linear discriminants. Stepwise linear discriminant analysis (S-LDA) was computed using the *klaR* package [53] in R-Studio Version 1.4.1717. Stepwise variable selection was set in both directions (forward and backwards), with a 0.01 least improvement of performance measure desired to include or exclude any variable. The number of variables to be included was not limited. Variable relative importance was calculated by dividing each variable increment in accuracy with the sum of the selected variables of increment in accuracy.

Random forests (RF) were computed using the *randomForest* package [54] in R-Studio Version 1.4.1717. The maximum number of trees was set to 500, which proved to be enough to stabilize the error of the analysis. Variable relative importance was calculated by dividing each variable Gini accuracy by the sum of all variables Gini accuracies.

All chemometric approaches generated confusion matrixes used for the evaluation of the multivariate statistical approach performance in correctly classifying the sample provenance according to their elemental profile.

Models’ performances in internal and external validation were evaluated in terms of accuracy (%), sensitivity (%), and specificity (%) according to [55]. Model overall accuracy was calculated by dividing the number of correctly classified samples by the total number of samples.

## 5. Conclusions

Stalked barnacles are among the most valuable crustacean resources from the Portuguese rocky shores, having a high socio-economic relevance allied to high market value, reinforcing the need to develop analytical techniques coupled with chemometric approaches that can efficiently trace the geographical origin of these seafood products. The high number of elements typically found in these filter feeder animals introduces a background noise in traditional non-selective multivariate approaches, such as CAP. The application of variable selection approaches such as S-LDA provided a high accuracy degree in classifying the samples according to their harvesting area while reducing the number of elements needed for this classification. Moreover, the selected elements are similar to the ones selected by other approaches, reinforcing the reliability of this selection. Simultaneously, and given the gastronomical role of stalked barnacles as delicacies, this approach also offers important information regarding the concentration of elements with recommended thresholds for human consumption, providing valuable insights into the food safety status of this seafood product. The development of the traceability chemometric tools, based on the untargeted elemental profiles of the edible peduncle of *P. pollicipes*, is a novel technique that allows to track the origin of this high-value commercialized and regulated food resource, thus providing relevant data regarding the capture effort of these organisms, aside from the information provided by the fishermen. Ultimately, this information can be used for updating and revising management schemes that allow maintaining a sustainable harvest of these organisms, while simultaneously informing the consumers about their safety and provenance.

## Figures and Tables

**Figure 1 molecules-27-01298-f001:**
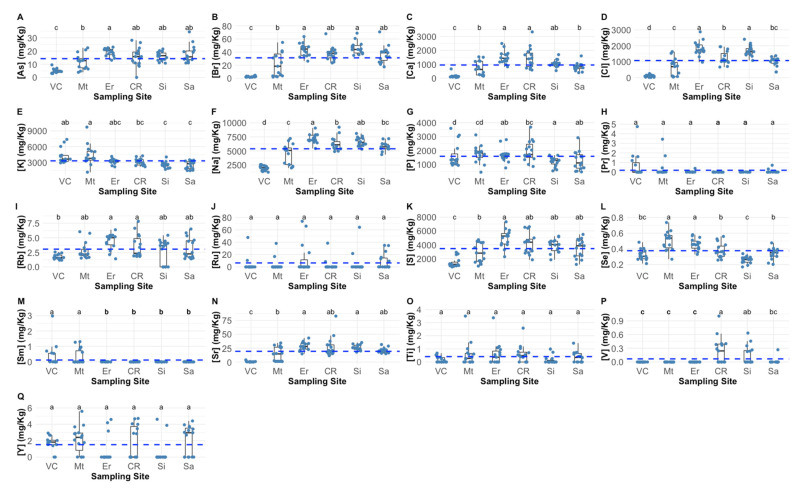
Arsenic, (**A**), bromine (**B**), calcium (**C**), chlorine (**D**), potassium (**E**), sodium (**F**), phosphorous (**G**), praseodymium (**H**), rubidium (**I**), ruthenium (**J**), sulphur (**K**), selenium (**L**), samarium (**M**), strontium (**N**), titanium (**O**), vanadium (**P**) and yttrium (**Q**) Pollicipes pollicipes peduncle concentrations (wet weight) of the individuals collected at the six sampling sites (VC—Viana do Castel; Mt —Matosinhos; Er—Ericeira; CR—Cabo Raso; Si—Sines; Sa—Sagres; dotted blue lines represent the average value detected for the whole dataset). Statistical letters denote significant differences between sampling sites at *p* < 0.05.

**Figure 2 molecules-27-01298-f002:**
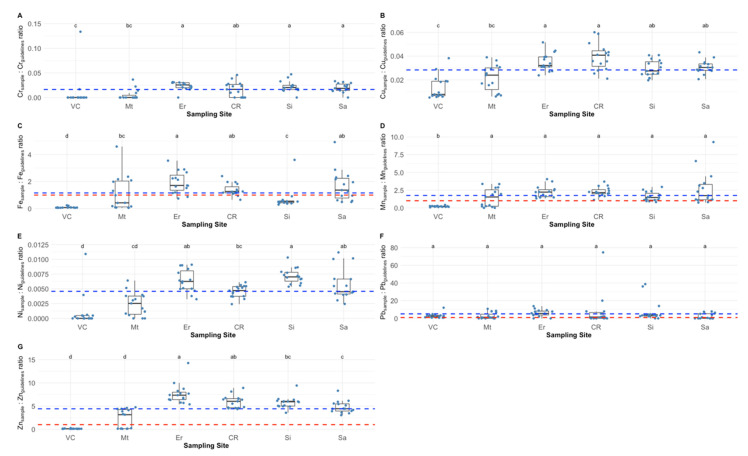
Chromium, (**A**), copper (**B**), iron (**C**), manganese (**D**), nickel (**E**), lead (**F**) and zinc (**G**) ratios between the measured values and the international thresholds in *Pollicipes pollicipes* peduncle, of the individuals collected at the six sampling sites (VC—Viana do Castelo; Mt—Matosinhos; Er—Ericeira; CR—Cabo Raso; Si—Sines; Sa—Sagres; dotted blue lines represent the average value detected for the whole dataset; the dotted red line represents the safety threshold ratio (ratio = 1) according to international regulatory authorities; for Cr, Cu and Ni the international threshold guidelines are far above the results attained from this study samples and thus the correspondent dotted red lines were are not displayed). Statistical letters denote significant differences between sampling sites at *p* < 0.05.

**Figure 3 molecules-27-01298-f003:**
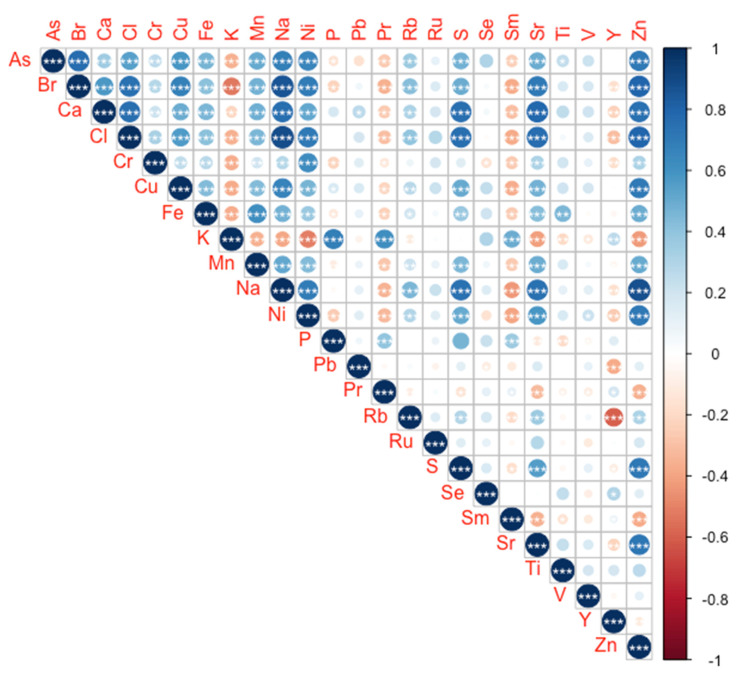
Spearman correlation matrix heatmap (ρ) between the elemental concentrations detected in Pollicipes pollicipes peduncle (asterisks denote significant correlations at *p* < 0.05 *, *p* < 0.01 ** and *p* < 0.001 ***).

**Figure 4 molecules-27-01298-f004:**
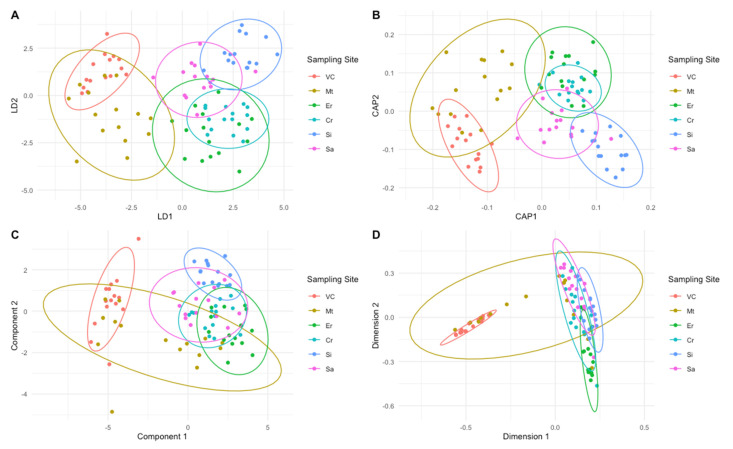
Linear discriminant analysis (LDA, **A**), canonical analysis of principal components (CAP, **B**), partial least-squares discriminant analysis (PLS-DA, **C**) and random forests (RF, **D**) projection plots of the Pollicipes pollicipes peduncle elemental profiles, from the individuals collected at Viana do Castelo (VC), Matosinhos (Mt), Ericeira (Er), Cabo Raso (Cr), Sines (Si) and Sagres (Sa) sampling sites.

**Figure 5 molecules-27-01298-f005:**
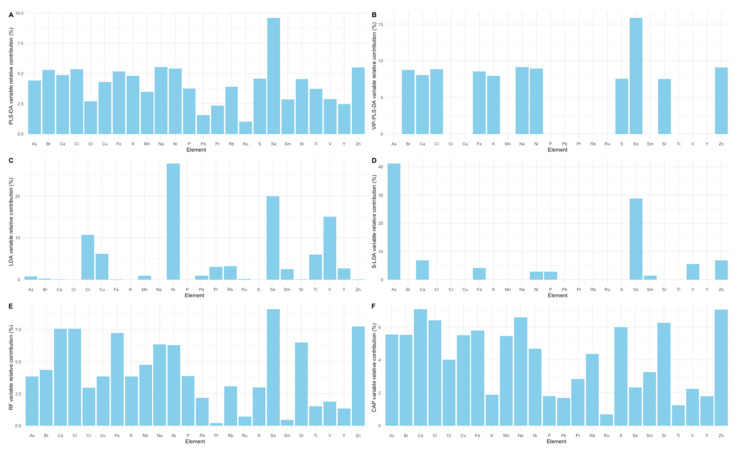
Element variable relative contribution in each of the chemometric models tested: Partial least-squares discriminant analysis (PLS-DA, **A**), variable importance in projection partial least-squares discriminant analysis (VIP-PLS-DA, **B**), linear discriminant analysis (LDA, **C**), stepwise linear discriminant analysis (S-LDA, **D**), random forests (RF, **E**) and canonical analysis of principal coordinates (CAP, **F**).

**Figure 6 molecules-27-01298-f006:**
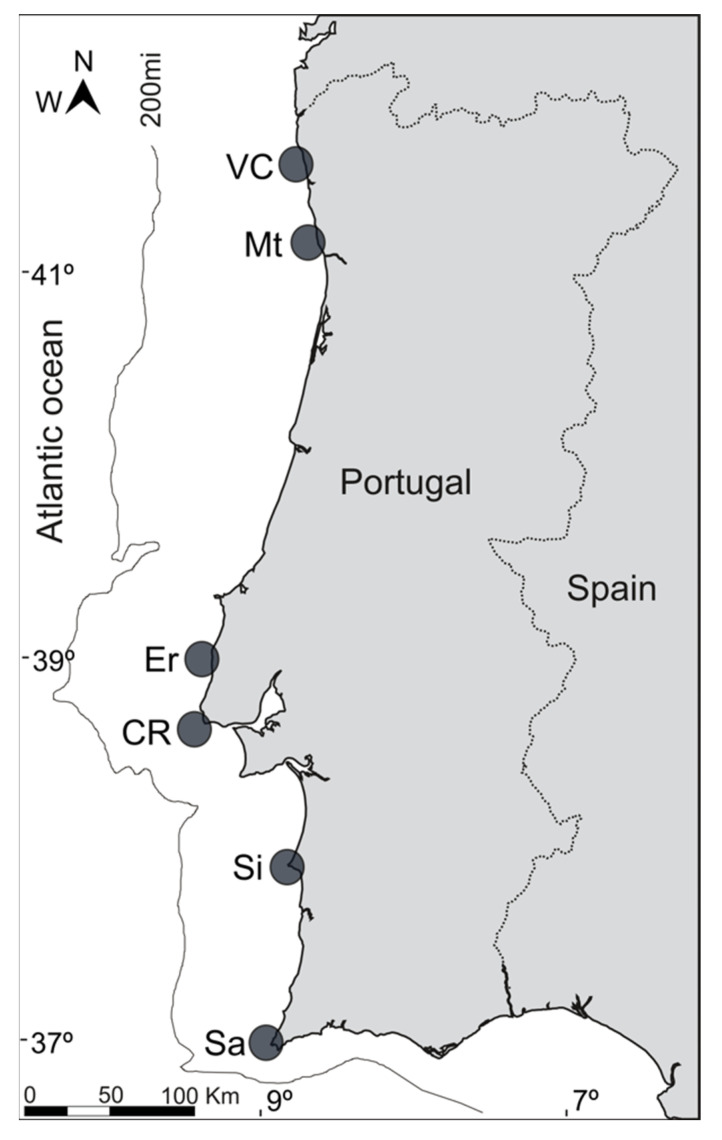
Location of the sampling sites of the specimens evaluated in the present study along the Portuguese Western coast (VC—Viana do Castelo; Mt—Matosinhos; Er—Ericeira; CR—Cabo Raso; Si—Sines; Sa—Sagres).

**Table 1 molecules-27-01298-t001:** Partial least-squares discriminant analysis (PLS-DA), variable importance in projection partial least-squares discriminant analysis (VIP-PLS-DA), linear discriminant analysis (LDA), stepwise linear discriminant analysis (S-LDA), random forests (RF) and canonical analysis of principal coordinates (CAP) confusion matrixes (correctly classified samples in bold) and model performance traits (overall model accuracy and group classification accuracy, precision, sensitivity and specificity), based on the elemental profiles of the individuals collected at Viana do Castelo (VC), Matosinhos (Mt), Ericeira (Er), Cabo Raso (Cr), Sines (Si) and Sagres (Sa) sampling sites.

		Predicted						Overall Accuracy (%)	Accuracy (%)	Precision (%)	Sensitivity (%)	Specificity (%)
Model	Origin	VC	Mt	Er	CR	Si	Sa					
**PLS-DA**	VC	**14**	1	0	0	0	0	77.8	90.9	70.0	93.3	90.3
	Mt	6	**6**	2	0	0	1		85.4	66.7	40.0	95.5
	Er	0	0	**14**	0	0	1		90.9	70.0	93.3	90.3
	CR	0	2	1	**11**	1	1		93.3	91.7	73.3	98.3
	Si	0	0	0	0	**15**	0		98.6	93.8	100.0	98.2
	Sa	0	2	1	1	1	**10**		89.7	76.9	66.7	95.2
**VIP-PLS-DA**	VC	**14**	1	0	0	0	0	77.8	90.9	70.0	93.3	90.3
	Mt	5	**10**	0	0	0	0		88.6	71.4	66.7	93.8
	Er	0	0	**13**	0	1	1		94.6	86.7	86.7	96.6
	CR	0	3	1	**8**	1	2		89.7	88.9	53.3	98.4
	Si	0	0	1	0	**14**	0		93.33	77.78	93.33	93.33
	Sa	1	0	0	1	2	**11**		90.9	78.6	73.3	95.2
**LDA (validation set)**	VC	**3**	0	0	0	0	0	72.7	91.4	50.0	100.0	90.6
	Mt	1	**4**	0	0	0	0		97.0	100.0	80.0	100.0
	Er	0	0	**6**	2	0	2		84.2	75.0	60.0	92.9
	CR	2	0	1	**7**	0	1		84.2	77.8	63.6	92.6
	Si	0	0	1	0	**7**	1		91.4	87.5	77.8	96.2
	Sa	0	0	0	0	1	**5**		86.5	55.6	83.3	87.1
**S-LDA**	VC	**14**	1	0	0	0	0	84.4	93.8	77.8	93.3	93.9
	Mt	4	**11**	0	0	0	0		93.8	91.6	73.3	98.5
	Er	0	0	**14**	0	1	0		93.8	77.8	93.3	93.9
	CR	0	0	3	**11**	0	1		93.8	91.6	73.3	98.5
	Si	0	0	0	0	**14**	1		96.2	87.4	93.3	96.9
	Sa	0	0	1	1	1	**12**		93.8	85.7	79.9	97.0
**RF**	VC	**12**	3	0	0	0	0	70.0	92.6	85.7	80.0	96.2
	Mt	2	**10**	2	0	0	1		87.5	71.4	66.7	93.0
	Er	0	0	**12**	0	1	2		88.7	70.6	80.0	91.1
	CR	0	1	2	**8**	2	2		86.3	72.7	53.3	94.8
	Si	0	0	0	0	**13**	2		88.7	68.4	86.7	89.3
	Sa	0	0	1	3	3	**8**		81.8	53.3	53.3	88.7
**CAP**	VC	**11**	3	0	0	0	1	73.3	88.0	68.8	73.3	91.7
	Mt	4	**10**	1	0	0	0		88.0	71.4	66.7	93.3
	Er	0	0	**12**	1	1	1		89.2	70.6	80.0	91.5
	CR	0	1	2	**11**	0	1		89.2	73.3	73.3	93.2
	Si	0	0	0	2	**12**	1		93.0	85.7	80.0	96.4
	Sa	1	0	2	1	1	**10**		88.0	71.4	66.7	93.3

**Table 2 molecules-27-01298-t002:** Mussel (ERM-CE278k) certified and analysed elemental values, uncertainty (mg/kg) and calculated extraction efficiency (average ± standard deviation, *N* = 5).

Element	Certified Value	Uncertainty	Measured Value	Extraction Efficiency (%)
Cr	0.73	0.22	0.67 ± 0.10	91.3 ± 5.3
Mn	4.88	0.24	3.35 ± 0.10	68.6 ± 1.9
Fe	161.0	8.0	198.03 ± 0.67	123.0 ± 0.4
Ni	0.69	0.15	0.78 ± 0.05	113.1 ± 5.8
Cu	5.98	0.27	7.10 ± 0.07	118,8 ± 1.0
Zn	71.0	4.0	73.29 ± 0.28	103.2 ± 0.4
As	6.7	0.4	7.25 ± 0.07	108.2 ± 0.9
Se	1.62	0.12	1.51 ± 0.03	93.3 ± 1.9
Rb	2.46	0.16	2.45 ± 0.05	99.6 ± 1.9
Sr	19.0	0.0	18.55 ± 0.34	97.6 ± 1.8
Cd	0.336	0.025	0.32 ± 0.02	96.6 ± 4.5
Pb	2.18	0.18	2.47 ± 0.05	113.3 ± 2.1

## Data Availability

The data presented in this study are available on request from the corresponding author.

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
