# Peer review of "Elemental Chemometrics as Tools to Depict Stalked Barnacle (Pollicipes pollicipes) Harvest Locations and Food Safety"

_molecules, 2022, doi:10.3390/molecules27041298_

Round 1
Reviewer 1 Report
This study explored the potential of total X-ray fluorescence spectroscopy based non-targeted multi-elemental fingerprinting of Pollicipes pollicipes peduncle using different chemometric approaches for geographical origin traceability assessment, with simultaneous evaluation of food safety values related to the presence and concentration of potentially toxic elements. This study has a practical significance and can be considered for publication in molecules. However, some improvements are needed prior to publication.
Although the significance of this study is discussed in an introduction, it is very important to discuss the novelty as well.
Please improve the quality of all the figures.
Which version of Primer 6 software is used for multivariate statistical analyses.
Conclusions should be amended to incorporate a broader discussion of the significance and potential application of this specific study.
Author Response
Lisbon, 07.02.2022
The authors would like to thank the reviewers for the opportunity to improve the manuscript with their useful comments and suggestions. All were considered and adequate corrections were performed as follow.
REVIEWER #1
Q: Although the significance of this study is discussed in an introduction, it is very important to discuss the novelty as well.
A: The text was revised, reinforcing the novelty of the present work as requested by the reviewer.
Q2. Please improve the quality of all the figures.
A: All images were enlarged as requested. Nevertheless, some resolution problems might occur due to pdf generation issues. Accordingly, the original figure files were uploaded separately in high resolution format, for final manuscript production.
Q3. Which version of Primer 6 software is used for multivariate statistical analyses.
A: The software version was added as requested.
Q: Conclusions should be amended to incorporate a broader discussion of the significance and potential application of this specific study.
A: Conclusion were amended in order to incorporate the reviewer’s suggestion.
Reviewer 2 Report
MOLECULES 1594646
Title: Elemental chemometrics as tools to depict stalked barnacle 2 (Pollicipes pollicipes) harvest locations and food safety
Comments of the reviewer
The manuscript is fascinating from a scientific point of view. However, some considerations by this reviewer must be evaluated by the authors before sending the final version.
Employing of the LDA:
LDA is the oldest classification method developed by Ronald Fisher. The initial assumptions are as follow:
- Each class is modeled by the normal distribution N (μk, Σk);
- Covariances matrices of the classes are identical: Σ1 = Σ2 = ... = Σk
- Did you verify these assumptions?
Using the Spearman correlation:
- Why did you use the Spearman correlation after using the Kruskal-Wallis and Bonferroni tests? Please, justify or use a non-parametric statistic tool.
Using PLS-DA and LDA for the multiclass problem:
1. Please, I recommend reading the article published by Brereton and Lloyd (DOI: 10.1002/cem.2609). The authors addressed very well the apllication of these chemometric tools for multiclass problem.
Supplementary material
- Why did you show results only for seven elements if you determined 24? Please, insert all of them in the supplementary material.
- Please correct the unit of concentration on the legend of the table.
- Can you justify the standard error is higher than the mean for a few elements?
Author Response
Lisbon, 07.02.2022
The authors would like to thank the reviewers for the opportunity to improve the manuscript with their useful comments and suggestions. All were considered and adequate corrections were performed as follow.
REVIEWER #2
Q: Employing of the LDA:
LDA is the oldest classification method developed by Ronald Fisher. The initial assumptions are as follow: Each class is modeled by the normal distribution N (μk, Σk); Covariances matrices of the classes are identical: Σ1 = Σ2 = ... = Σk. Did you verify these assumptions?
A: The authors would like to thank the reviewer for raising this question, and yes both assumptions were verified, as part of script and functions used. For clarity, this information was added to the manuscript.
Q: Using the Spearman correlation: Why did you use the Spearman correlation after using the Kruskal-Wallis and Bonferroni tests? Please, justify or use a non-parametric statistic tool.
A: Spearman correlation test is a non-parametric test (https://statistics.laerd.com/statistical-guides/spearmans-rank-order-correlation-statistical-guide.php). While Kruskal-Wallis tests were used to preform pairwise comparisons between groups of two dataset groups, Spearman correlation tests allow to evaluate how the variables behave and correlate with each other using the whole dataset (and not between groups). Thus, both approaches are non-parametric and complementary.
Q: Using PLS-DA and LDA for the multiclass problem: Please, I recommend reading the article published by Brereton and Lloyd (DOI: 10.1002/cem.2609). The authors addressed very well the apllication of these chemometric tools for multiclass problem.
A: The authors would like to thank the reviewer for the reading suggestion. A disclaimer regarding possible interferences from the PLS-DA was added. Nevertheless, the authors have reasons to believe that the disadvantages pointed out by Brereton and Lloyd, are not influencing the conclusions in the present work, since the PLS-DA results are perfectly in line with the results attained from the other chemometric approaches, not only in terms of classification results, but also in terms of the variable importance attributed by PLS-DA and by the other approaches used.
Q: Supplementary material. Why did you show results only for seven elements if you determined 24? Please, insert all of them in the supplementary material. Please correct the unit of concentration on the legend of the table. Can you justify the standard error is higher than the mean for a few elements?
A: All elements were added as requested. Still, we wish to clarify that in the previous version only the elements that were not presented in the figures of the manuscript (either as concentration or as ratio to reference values) were included in the supplementary table, to avoid data duplication. Regarding the unit of concentration, the one presented is the correct one, as it can also be observed in the boxplots presented in the manuscript (mg/Kg ww, wet weight). Regarding the high standard error in some elements this can be due to a multitude of reasons that derive mostly from the natural variation of environmental samples, especially biological matrices.
Round 2
Reviewer 1 Report
Manuscript is well revised and can be considered for publication in molecules.